# Network Theory Analysis of Allosteric Drug-Rescue Mechanisms in the Tumor Suppressor Protein p53 Y220C Mutant

**DOI:** 10.3390/ijms26146884

**Published:** 2025-07-17

**Authors:** Benjamin S. Cowan, Kelly M. Thayer

**Affiliations:** 1Department of Computer Science, Wesleyan University, Middletown, CT 06457, USA; bcowan@wesleyan.edu; 2College of Integrative Sciences, Wesleyan University, Middletown, CT 06457, USA; 3Department of Chemistry, Wesleyan University, Middletown, CT 06457, USA

**Keywords:** networks, kernel, transformations, distance, metrics, allostery, protein, electrostatics, simulation, p53

## Abstract

Network theory analysis has emerged as a powerful approach for investigating the complex behavior of dynamic and interactive systems, including proteomic systems. One key application of these methods is the study of long-range signaling dynamics in proteins, a phenomenon known as allostery. In this study, we applied computational models using network theory analysis to explore long-range electrostatic interactions and allosteric drug rescue mechanisms in the DNA-binding domain (DBD) of the p53 protein, a critical tumor suppressor whose dysfunction, often caused by missense mutations, is implicated in over 50% of human cancers. Using heat kernel and Wasserstein distance-based analyses, we explored the allosteric behavior of p53-DBD constructs with the Y220C mutation in the presence or absence of allosteric effector drugs. Our results demonstrated that these network theory-based protocols effectively detected the differential efficacies of small molecule allosteric effector drug compounds in restoring long-range electrostatic dynamics in the Y220C mutant. Furthermore, our approach identified key long-range electrostatic interactions critical to both the nominal and drug-rescued functionality of the p53-DBD, providing valuable insights into allosteric modulation and its therapeutic potential.

## 1. Introduction

The p53 protein, often referred to as the “Guardian of the Genome” [1,2], plays a central role in tumor suppression and transcription regulation. As a transcription factor, p53 regulates the expression of genes critical for numerous cellular processes, including cell division, differentiation, metabolism, DNA repair, apoptosis, and genomic stability. These functions position p53 as a master regulator of cell fate [3] and genomic stability in unstressed cells [4,5]. Structurally, p53 comprises 393 amino acids organized into several distinct functional domains. The N-terminal region includes two transactivation domains (TAD1 and TAD2), which play roles in directing p53 target gene selection and transcriptional activation [6,7]. A proline-rich domain (PRD) follows, contributing to p53’s transcriptional activity [8] and structural stability [9]. The central DNA-binding domain (DBD), a highly conserved region [10], facilitates sequence-specific DNA binding essential for p53’s function as a transcription factor [11]. The DBD forms its interaction surface using a loop-sheet-helix motif stabilized by zinc coordination. The C-terminal to the DBD is the tetramerization domain (TD), which enables p53 to form a functional tetramer, and the intrinsically disordered C-terminal regulatory domain (CTD), which inhibits DNA-binding activity until specific post-translational modifications relieve this inhibition.

Dysfunction of p53 is directly implicated in over 50% of human cancers, with the remainder often involving disruptions in pathways regulated by p53 [12]. Approximately 75% of cancer-associated p53 mutations are missense mutations [13], with over 97% of these occurring in the DBD [14,15]. These mutations often lead to structural destabilization and loss of function, conferring a selective advantage to cancer cells by inhibiting the tumor-suppressing activities of p53 [13]. Among these, the Y220C mutation is the most common missense mutation outside the DNA-binding surface, making it a key model for investigating allosteric modulation [16]. The significant prevalence of p53 mutations and their central role in tumorigenesis have made mutant p53 a prominent target for anti-cancer therapeutics.

Allosteric regulation offers a promising strategy for restoring mutant p53 function. Originally described by Monod et al. [17], allostery involves long-range coupling between distant binding sites and a protein’s active site through structural and energetic changes. Recent research expands this concept to investigate shifts in the energetic dynamics of electrostatics across residue interaction networks [18,19,20]. Energetic models of protein allostery emphasize the global role of long-range energetic molecular interactions, initiated by new contacts at sites distal to the protein’s catalytic domain, in reshaping the protein’s global energetic landscape [21]. Such long-range energetic changes reorganize residue-residue interactions at the catalytic site, resulting in functional transitions of the protein [21]. For p53, allostery is particularly valuable because its function relies on an unobstructed active site for DNA binding. Small-molecule targeting allosteric sites could stabilize the DNA-binding domain (DBD) and enhance DNA affinity without interfering directly with the active site.

Energetic interactions play a crucial role in protein function [22], stability [23], catalysis [24], and allostery [25]. Electrostatic networks facilitate long-range communication between residues, enabling allosteric signaling. In p53, understanding how these electrostatic interactions influence DNA binding and stability is key to developing effective allosteric drugs. Studies show that electrostatic interaction patterns in the p53-DBD are critical for allosteric control and DNA-binding affinity [26]. Additionally, recent studies have shown that in model allosteric systems the energetic binding contributions of electrostatic network interactions are a more significant determinant of binding affinity than other forms of energetic dynamics such as van der Waals [27] with more recent studies of p53-DNA binding demonstrating similar conclusions [28]. Allosteric mutations such as Y220C disrupt these networks, causing loss of function, but second-site suppressor mutations can partially restore activity. This highlights the potential of long-range energetic modulation as a therapeutic strategy.

In this study, we employ molecular dynamics (MD) simulations and network-theoretic analyses to demonstrate the utility of a novel protocol for identifying long-range energetic changes involved in allosteric small molecule binding on p53-DBD functional rescue, focusing on the Y220C mutation as a model system. We apply our energetic network theoretic methodologies to the MD simulation data of seven p53 constructs: the wild-type p53 protein, the Y220C-mutant p53 protein, the Y220C-mutant p53 bound to the Y220C mutant allosteric reactivator drug compound “**PK11000**” [29], and four p53 protein constructs bound to a panel of small molecule effector compounds recently reported by a MD-based docking study performed by Han et al. [30] (Figure 1). These small molecule effector compounds were selected specifically because of their demonstrated efficacy in rescuing the dynamics and function of mutant p53 in prior studies [13].

Many network-theoretic techniques have been created to investigate the role of residue networks in protein function [31,32], structural flexibility [33], and protein stability [34]. In order to perform network-theoretic analyses to study proteins, data from molecular dynamics (MD) simulations of proteins are utilized since MD simulations can yield an ensemble of protein conformations that capture both the backbone and side-chain level differences in molecular interactions [35]. Rather than protein structure networks (PSNs) and contact networks, which merely encode edge presence between nodes by a specified geometric distance threshold, we construct locally thresholded electrostatic interaction networks [30,36], which embed residues (nodes) connected by weighted edges representing the level of electrostatic interaction between residues. Such a methodology enables investigations into longer-range forms of residue connectivity beyond covalent-bond distance thresholds alone [37].

Utilizing vector transformations to project higher-dimensional patterns into lower-dimensional spaces offers utility to investigate the often complex long-range dynamical properties inherent to network dynamics. Kernels embody one such transformation procedure, and the heat kernel lends itself to this study to capture the organization of residue’s long-range electrostatic network dynamics across simulation time [27]. By utilizing kernel-based network transformations and distance metrics, this methodology can characterize residue-level differences in long-range electrostatic interactions across simulations of wild-type, Y220C mutant, and small molecule-complexed states of p53. Our study provides insight into the important role of electrostatics in p53 allostery and the differential efficacies of allosteric drugs, including the known p53-DBD allosteric reactivator PK11000 [29] and other small molecules involved in functional rescue of p53. Such findings we hope will offer a foundation for the development of innovative cancer therapeutics targeting this critical tumor suppressor protein.

## 2. Results

### 2.1. Energetic Network Generation and Electrostatic Heat Kernel PCA Projection

To capture the time evolution of long-range energetic dynamics in the p53-DBD, energetic interaction networks were computed for each of 152 regularly sampled frames of a p53-DBD construct’s MD simulation. Each sampled frame’s energetic network captured the corresponding interaction energy value between each pair of the p53-DBD’s 193 amino acids. The details of the trajectory sampling and network generation processes are provided in the Methods Section. The energetic interaction networks were generated for the two unbound p53-DBD constructs: wild type (WT) and the Y220C-mutant (Y220C-ub), the Y220C-mutant bound to the known p53 allosteric reactivator PK11000 (Y220C-PK11000), and the Y220C-mutant bound to four additional small-molecule allosteric effector compounds (**Y220C-8**, **Y220C-20**, **Y220C-22**, and **Y220C-27**).

Energetic networks were generated for both electrostatic and Van der Waals (VDW) interactions. However, VDW edge weights across all seven constructs were consistently near zero in value, aligning with expectations that VDW forces predominantly affected short-range, localized interactions [38]. Due to the insignificant edge weights for the VDW networks, our subsequent analyses focused exclusively on the constructs’ generated electrostatic networks, which were more relevant for investigating long-range energetic dynamics in the DBD. Complete data on the interaction energy edge weights for both the electrostatic and the VDW networks can be found in the supporting materials.

For each construct, each of its 152 electrostatic networks was transformed into a corresponding heat kernel matrix and projected into a shared R^3^ embedding space across the first three principal components (PC^1^, PC^2^, and PC^3^) of the mean-centered heat kernel. The resulting embeddings, visualized in Figure 1a,b, represented the electrostatic associations of the p53-DBD’s 193 residues (excluding C182 and C220) across the 152 sampled frames of simulation. Each of the construct’s 193 residues thus had 152 representative node embeddings in the shared R^3^ PC space reflecting its electrostatic covariance with all other p53-DBD residues across simulation time. Closer proximity between residue embeddings indicated higher covariation and stronger electrostatic connectivity.

The R^3^ heat kernel projections revealed key trends in electrostatic connectivity across the seven investigated p53-DBD constructs. Residues with higher numbers of electrostatic interactions were reflected by node embeddings with values ≥ 0.0105 along PC^1^ (Figure 2). These embeddings’ higher values correlated to more “hub-like” behavior in organizing the network’s organization of energetic connectivity. In contrast, residues with lower connectivity (heat kernel values ≤ 0.0095) had a more peripheral role in the structuration of constructs’ energetics. Shifts in the distribution of node embeddings between the constructs indicated how Y220C mutation and/or the presence of allosteric drugs modulated the global energetic landscape of the p53-DBD. While changes were evident across all three principal components, most positional variations occurred along PC^1^ and PC^2^, justifying the simplified visualization of the embedded networks across two dimensions in Figure 2a,b. These findings highlight that the Y220C mutation and allosteric drug binding influenced p53-DBD’s electrostatic network organization. By modulating residue connectivity, these factors reshaped the protein’s global energetic landscape, providing insights into the mechanisms of functional disruption and potential rescue.

### 2.2. Embedding Error Analyses

To assess the impact of allosteric drug binding and the Y220C mutation on p53-DBD dynamics, we performed Wasserstein embedding error analyses on the electrostatic heat kernel projections in PC space. This analysis captured how the organization of each residue within the DBD’s energetic network varied between pairs of the seven investigated p53-DBD constructs, providing a comparative view of significant reorganizations in the electrostatic dynamics between wild-type, unbound Y220C-mutant, and drug-bound Y220C p53-DBD constructs. The embedding error (EE) value of each residue quantifies the degree of positional variance in its 152 R^3^ embeddings between two constructs. We denote this residue-wise distribution of EE values calculated between pairs of constructs *U* and *V* as *U_V*.

In the first set of six EE distributions between the wild-type construct and each of the six possible Y220C mutant p53-DBDconstructs (Figure 3a), residues with high EE values were predominantly localized in loop and helical motifs of the p53-DBD. Residues in the EE distribution demonstrated prominently higher EE values between the wild-type and unbound Y220C mutant construct, WT_Y220C-ub across the L1 loop (residues L114-K120), L6 loop (residues P222-C229), and L3 loop (residues N247-R249). In contrast, the EE distributions involving the wild type and each of the five Y220C mutant drug-bound constructs (WT_Y220C-8, WT_Y220C-22, and WT_Y220C-PK11000) exhibited lower EE values in these corresponding L1, L6, and L3 loop regions.

A second set of five EE distributions between the unbound Y220C mutant and each of the mutant effector drug-bound constructs (Y220C-8, Y220C-20, Y220C-22, Y220C-27, and Y220C-PK11000) revealed regions of high EE value overlapping with those found in the distributions between the wild type and the unbound Y220C mutant (Figure 3b). High EE values were consistently identified in the S6-S7 loop (residues N207-R211), particularly at R209. Among the drug-bound constructions, Y220C-22 displayed the highest embedding error values in this region, while Y220C-8 and Y220C-PK11000 exhibited comparatively moderate values, suggesting differential effects of the drug compounds on the electrostatic dynamics of residues in this loop region.

To further quantify the extent of electrostatic divergence and restoration, minimal embedding error analysis (MinEEa) and maximal embedding error analysis (MaxEEa) were performed.

MinEEa (Figure 4a,b) identified residues with the smallest embedding error values across the distributions between the wild type and each of the six Y220C constructors investigated, highlighting regions where drug binding restored wild-type-like dynamics (Figure 4a). The results, shown in Figure 4b, revealed that the Y220C-PK11000 construct accounted for the largest proportion (41.45%) of residues with minimal embedding error values, particularly in regions spanning S3 (residues C141-W146), S4 (residues T155-Y163), L2 (residues S183-L194), and L6 (residues E221-T230). Constructs Y220C-22 and Y220C-8 ranked second and third, with proportions of 24.35% and 21.76%, respectively. Conversely, the unbound Y220C construct exhibited the smallest proportion of residues with minimal embedding errors (1.55%), followed by Y220C-27 (4.66%) and Y220C-20 (6.22%).

In MaxEEa (Figure 4c,d), the residues with the largest embedding error values were identified across the EE distributions between the unbound Y220C mutant DVD construct and the five Y220C mutant drug-bound constructs. MaxEEa highlighted regions where electrostatic dynamics in the mutant drug-bound constructs were most divergent from those in the unbound Y220C mutant DBD construct. The constructs Y220C-22, Y220C-PK11000, and Y220C-8 exhibited the highest proportions of maximal embedding error values, particularly in regions such as the L1, L3, and the H2 helix. In contrast, Y220C-27 and Y220C-20 showed the lowest proportions, indicating reduced electrostatic divergence from the unbound Y220C construct (Figure 4d).

To better understand the directional redistribution of residue embeddings, embedding error difference (EED) analysis was performed (Figure 5).

For each of the five drug-bound Y220C constructs, EED calculated the difference between each residue’s embedding error values calculated relative to the wild-type construct and the unbound Y220C construct. Residues with positive EED values exhibited heat kernel embeddings in each Y220C-mutant drug-bound construct closer to their heat kernel embeddings in the R^3^ PC space in the wild-type construct, while those with negative values shifted toward heat kernel embeddings more similar to their embeddings in the R^3^ PC space in the unbound mutant Y220C construct. As such, the restorative impact of drug binding on Y220C-mutant network organization toward the residue’s nominal wild-type long-range electrostatic interactions was operationalized as residues displaying positive EED values. Contrastingly, residues displaying negative EED values for each of the Y220C drug-bound constructs demonstrated the effect of that drug compound on destabilizing that residue’s long-range electrostatic network dynamics toward the unbound Y220C mutant construct.

Positive EED values were consistently observed in regions spanning S3 (residues 140–144), S5-S6 (residues E198-R202), and L6 (residues E221-T230) across all five drug-bound constructs. Notably, residues in the S6-S7 loop displayed distinct EED trends among the drugs. Y220C-22 resulted in highly positive EED values in this region, while Y220C-PK11000 and Y220C-8 led to more negative values, particularly at R209. Additionally, residues proximal to the mutation site, including L1, S2-S2′, and the H2 helix, exhibited significant positive EED values in constructs bound to PK11000, 8, and 22. Constructs bound to 20 and 27 displayed predominantly negative EED values, suggesting a lesser restorative effect on p53-DBD dynamics. Overall, these analyses demonstrated that allosteric drug binding significantly modulated electrostatic network dynamics in the p53-DBD, with certain compounds, such as **PK11000**, **8**, and **22**, exhibiting stronger restorative effects toward wild-type behavior.

To investigate the role of allosteric rescue in regions critical for DNA binding and overall p53-DBD function, we analyzed the distribution of residues with high EED values across key loop and helical motifs, including L1, L2, H1, S6-S7, L6, L3, and H2. These regions were selected due to their functional implications in allosteric signaling and their involvement in stabilizing DNA interactions [39]. Table 1 summarizes the proportions of residues within these regions exhibiting increasingly positive EED values, and Figure 6a projects these residues with significant EED values onto the p53-DBD structure.

Among the regions analyzed, S6-S7 and L6 demonstrated the highest proportions of residues with EED values ≥ 0.0005, underscoring their significance in allosteric rescue. L1 and H2 also showed a higher percentage of residues with positive EED values compared with other regions, such as H1, L2, and L3.

To compare the efficacy of the five drug compounds, we calculated the average EED values for residues within these seven key loop regions under each Y220C drug-bound condition (Figure 6b).

The analysis revealed that L6, followed by S6-S7 and L1, demonstrated the most pronounced responses to drug binding. Among the drugs, PK11000, compound **22**, and compound **8** were the most effective in redistributing electrostatic dynamics in these regions toward wild-type-like behavior. The only effector compound capable of reestablishing nominal electrostatics across L6, S6-S7, and L1 simultaneously was compound **22**, demonstrating its comparatively significant efficacy for redistributing the global electrostatics of the Y220C mutant p53-DBD to wild-type-like dynamics.

## 3. Discussion

The population shift model of protein dynamics suggests that distal allosteric perturbations can modulate the local functional dynamics of active sites by reorganizing the global energetic landscape of a protein. This mechanism alters the equilibrium of functional ensemble states, as seen in p53-DBD, where mutations such as Y220C shift residue interactions toward non-functional conformations, resulting in loss of DNA-binding affinity. Modulation of the global activity of a protein elicited by allosteric perturbation leads to redistributions across each residue’s dominant inter-residue interactions and the functional states the entire protein can occupy [40]. This study employed novel computational network analyses to investigate how allosteric drugs influence the long-range electrostatic dynamics of p53-DBD and restore functional behavior disrupted by the Y220C mutation. By leveraging Wasserstein distance-based embedding error (EE) metrics, we quantified the extent and distribution of significant energetic changes induced by allosteric drugs, evaluating their differential efficacies in rescuing p53-DBD functionality.

### 3.1. Heat Kernel Analyses Capture Dominant Energetic Connections Across Constructs

The heat kernel methodology was instrumental in identifying the global networked dynamics of long-range electrostatic interactions across wild-type, Y220C mutant, and drug-bound p53-DBD constructs. Optimized for embedding electrostatic networks into R^3^ space, the heat kernel projections effectively partitioned residues based on their connectivity to other residues, capturing the organization of each construct’s electrostatic networks over simulation time. Residues with high connectivity, particularly arginine residues distributed throughout the p53-DBD, consistently separated along the principal component axes due to their numerous high-intensity interactions. Arginine residues are essential for stabilizing energetic pathways, diffusing thermal perturbations, and maintaining global electrostatic interactions critical to p53 function [41,42]. The ability of the heat kernel embeddings to encode long-range energetic relationships, beyond mere structural proximity, underscores their value in elucidating the global and local electrostatic dynamics of p53-DBD.

### 3.2. Wasserstein Embedding Error Analyses Reveal Long-Range Reorganizations

Allosteric mutations operate by modulating the dynamics of significant fractions of residues across globular protein structure, deviating long-range interactions between functional domains away from their native stabilities that facilitate the entire protein functionality [43]. An allosteric framework for protein functional rescue emphasizes how introduced interactions at sites distal to functional domains reestablish active site functionality by recovering significant long-range stabilities lost from the effects of mutational disruption. Our embedding error analyses provided insights into how the introduction of small molecule compounds at sites allosteric to the p53-DBD modulate electrostatic network interactions within the Y220C mutant. By embedding each residue’s long-range electrostatic dynamics in the latent PC space, we linked local residue-level changes to global reorganizations in p53-DBD function. Comparisons of embedding error distributions showed that unbound Y220C p53 exhibited the largest deviation from wild-type-like electrostatic interactions, particularly in loop and helical motifs critical for DNA binding and stability [44,45]. Drug binding partially restored these interactions, as evidenced by reduced embedding error values in key regions of the DBD.

MinEEa revealed that PK11000 accounted for the largest proportion (41.45%) of residues displaying wild-type-like electrostatic interactions, followed by drugs 22 and 8. MaxEEa further demonstrated that drug compound **22** induced the most significant redistribution of residues away from Y220C mutant-like electrostatic dynamics, with compounds **8** and **PK11000** also showing substantial effects. These findings highlight the ability of these compounds to stabilize residue-level interactions across the DBD, promoting a shift toward functional conformations.

### 3.3. Allosteric Disruption and Rescue in p53-DBD Dynamics

The globally disruptive nature of the Y220C mutation was evident in the unbound construct’s low proportion of residues with wild-type-like electrostatic dynamics compared with any of the drug-bound constructs. Notably, regions of high embedding error between wild type and Y220C-ub (e.g., L1, L6, S6-S7, and H2) overlapped with high embedding error regions in comparisons of Y220C-ub and drug-bound constructs. This suggests that the allosteric effects of the Y220C mutation are closely mirrored by the redistribution effects of drug binding. Residues proximal to Y220C (e.g., L6) and distal regions such as L1 and H2, which are critical for DNA-binding and structural stability, exhibited significant recovery in electrostatic interactions upon drug binding. The EED analysis quantified the residue-level impact of each drug, identifying regions where electrostatic interactions shifted toward wild-type-like dynamics. Across all five compounds, residues in L6 consistently exhibited the greatest degree of rescue. Regions such as the S2-S2′ hairpin and H2 helix, which are directly involved in DNA complex formation [46], also displayed positive EED values for multiple Y220C drug-bound DBD constructs, evidencing the role of allosteric compounds’ capacities to rescue active-site interactions from allosteric mutation. Residues of the S6-S7 loop region demonstrated substantial variability in EED value dependent on the specific drug-bound construct, with Y220C-22 showing the most positive EED values followed by Y220C-27. Conversely, Y220C constructs bound to compounds **8** and **PK11000** produced minimal recovery of this region. Recent mutational studies have demonstrated residues of the S6-S7 region, particularly R209, to be crucial for stabilizing energetically favorable states of the p53-DBD [47], regulating p53 function despite its distal location relative to the DNA binding interface at H2 with mutation of R209 being directly implicated in several cancers [48]. Our findings may thus evidence a highly sensitive role of electrostatic network organization of S6-S7 loop residues, implicated in long-range signaling between L6 and the DNA-binding L1 loop [44], for both regulating p53-DBD function and being a key target for allosteric drug rescue [39,49]. The overlap of high EED regions with functional interfaces highlights the utility of electrostatic network analyses in identifying critical sites for therapeutic targeting.

### 3.4. Regional Analysis of Allosteric Rescue in p53-DBD

To investigate the role of allosteric rescue in regions critical for DNA binding and overall p53-DBD function, we analyzed the distribution of residues with high embedding error difference (EED) values across key loop and helical motifs, including L1, L2, H1, S6-S7, L6, L3, and H2. These regions were selected due to their functional implications in allosteric signaling and their involvement in stabilizing DNA interactions [39]. L1 and H2 also showed a higher percentage of residues with positive EED values compared with other regions, such as H1, L2, and L3. This is particularly striking given that L1 and L6 are among the most distal regions from the Y220C mutation site. These findings suggest that allosteric drug binding facilitates coordinated electrostatic network reorganizations that span the DBD, effectively coupling proximal regions like L6 and S6-S7 with distal regions like L1 and H2, which directly interact with DNA.

### 3.5. Mechanistic Insights into Allosteric Rescue

One of the main goals of the present study was to uncover the underlying signaling pathways through which rescue compounds restore function to the Y220C mutant of p53. By using heat kernel-based network analysis, we identified shifts in long-range communication patterns that propagated from the ligand-binding site toward functional regions of the protein. These shifts were not evident from structural data alone but were captured in the heat kernel embeddings of electrostatic network reorganizations. Our results from the EED analysis suggested that the rescue mechanism operated by re-establishing disrupted electrostatic pathways, enabling more native-like residue–residue interactions that are critical for restoring wild-type behavior.

Across the investigated motifs of the p53-DBD, regions demonstrating the most dramatic “response” to the presence of the five effector compounds appeared to be L6, followed by S6-S7 and L1. As shown by our studies of the average EED value in Figure 6b, the L1and L6 motifs demonstrated the greatest electrostatic network reorganizations as dependent on the complexing of the allosteric compounds investigated with Y220C mutant p53-DBD. Out of the five compounds investigated, only compound **22** led to an average positive EED value across all three of these motifs. These findings align with previous studies conducted by the Thayer lab, which used MD-MSM and MD-Sector analyses to demonstrate that PK11000 and compound **22** best recapitulate wild-type dynamics in regions L6 and L1, where Y220C and DNA-binding residue K120 reside, respectively [13]. Our current results provide an additional layer of validation by demonstrating the electrostatic basis for these observations. Specifically, the coordinated rescue of electrostatic networks across L6, S6-S7, and L1 by compound **22** highlights the importance of these regions in stabilizing p53-DBD dynamics and DNA-binding function.

Further support for the role of electrostatics in allosteric rescue comes from our observation that 32% of residues with high EED values are located in the beta-sheet core of the p53-DBD, particularly in structurally adjacent sheets (e.g., S10 with S9, S4, and S8 with S5). The simultaneous rescue of residues in the structurally stable core and more dynamic loop regions indicates that allosteric drug effects are not solely mediated through conformational changes but are also driven by long-range electrostatic interactions. This supports the hypothesis that energetic reorganizations play a pivotal role in restoring nominal p53 function [28]. Our results therefore can provide a quantitative means of ascertaining in p53-DBD purely theoretical propositions of allosteric networks via redistributions through protein energetics [18].

### 3.6. Implications for Drug Design and Therapeutic Development

Our findings reinforced the therapeutic potential of the known p53-reactivator PK11000 and compound **22** as allosteric effectors capable of restoring wild-type-like electrostatic dynamics in p53-DBD. The Y220C mutant p53-DBD construct bound to compound **22** yielded the second-highest proportion of residues with minimal embedding errors in MinEEa (second only to the Y220C-PK11000 construct) and the highest proportion with maximal embedding errors in MaxEEa, underscoring its efficacy in restoring wild-type-like behavior across the DBD. Moreover, the ability of compound **22** to redistribute electrostatic dynamics in regions such as L6, S6-S7, and L1 highlights its potential to address the deleterious effects of the Y220C mutation through a coordinated allosteric mechanism. Recent evidence has pointed toward the dynamic coupling between both the L1 and S6-S7 loops in enabling DNA binding to exert long-range allosteric effects through H2 on DBD functioning [47,50], with other studies demonstrating disruptions in long-range motional correlations between L1 and L6 [44] consequent of Y220C mutation to the DBD. Compound **22**’s reversible nature of binding further enhances its viability as a therapeutic agent, distinguishing it from the PK11000, which, while initially designed to noncovalently interact with p53-DBD, covalently modifies cysteine residues 182 and 277 in the DBD [29]. PK11000’s unexpected covalent modifications have thus led to concerns over its potential off-target effects and toxicity to p53, making it unsafe for clinical use [13]. In contrast, compound **22** demonstrates a comparable efficacy in rescuing important long-range electrostatic interactions across the DBD, including those of S6-S7, while reversibly binding at an allosteric site. This characteristic corroborates recent findings of compound **22** as a particularly promising candidate for further development as a safer therapeutic alternative to PK11000 [13].

Our approach can be extended to study additional hot spot mutations in p53, many of which are of high biological and clinical interest due to their prevalence in. In order of respective prevalence in p53-DBD, R175H, R248Q, R273H, R248W, R175L, R273C, R282W, R248L, and R175P are of particular interest, with the first six of these mutations being accountable for 30% of all p53-DBD oncogenic mutations [51]. More broadly, the network-based framework we present is readily generalizable to any residue of interest in systems for which MD simulations are available. We view this method as a flexible tool with broad applicability for uncovering dynamic and allosteric consequences of mutations across a range of protein systems.

## 4. Materials and Methods

### 4.1. MD Simulation Trajectory Specifications for p53-DBD Constructs

The MD simulation trajectory data for the unbound wild-type p53-DBD construct, the unbound Y220C p53-DBD mutant construct, and the five mutant Y220C p53-DBD constructs bound to effector compounds (**PK11000**, **8**, **20**, **22**, and **27**) were sourced from Han et al.’s recently published MD docking and MD-MSM study on the Y220C p53 mutant [30].

All-atom MD simulations for each p53-DBD construct (residues 96–290) were conducted for 1 microsecond (10,000 simulation frames) using explicit solvent models, water, DNA, and counterions [30]. Simulations followed a standard protocol developed by the Thayer lab, employing the AMBER14 and AMBER16 packages with the AMBERTOOLS14 (v14, University of California, San Francisco, CA, USA) suite [52,53,54]. The FF19SB force field [54,55] was used for proteins, and the TIP3P potential was applied for solvent modeling.

Initial configurations of the p53-DBD wild-type (PDB ID: 1TUP) and Y220C+PK11000 (PDB ID: 5LAP) constructs were obtained from the Protein Data Bank (PDB). Residue Y220 was mutated to C220 using PyMOL (v1.8, Schrodinger, New York, NY, USA) [56]. PK11000 was parameterized using the Generalized Amber Force Field (GAFF) [30,57], implemented via Antechamber in the AMBER suite (University of California, v.16, San Francisco, CA, USA), [58] and covalently bonded to C182 using the t-Leap program suite (v.16, University of California, San Francisco, CA, USA) [52]. For the effector-bound constructs, docking of compounds **8**, **20**, **22**, and **27** to the average structure of the Y220C mutant was performed using AutoDock Vina (v1.20, Scripps Research, San Diego, CA, USA) [59,60], with zinc coordination parameterized using the Zinc AMBER Force Field (ZAFF) [61].

Each system was neutralized with Na+ counterions and treated with particle mesh Ewald periodic boundary conditions using a 10 Å Lennard–Jones cutoff in a truncated octahedral box [62]. Hydrogen bond motions were constrained using the SHAKE algorithm [63]. Systems were energy-minimized with progressively reduced solute constraints, heated to 300 K, and equilibrated using the Berendsen thermostat [64]. MD simulations were performed using the parallelized CUDA (v12.0, NVIDIA, Santa Clara, CA, USA) version of the pmemd routine [65] on NVIDIA GPUs (NVIDIA, Santa Clara, CA, USA). RMSD convergence analyses were conducted using the cpptraj utility in AMBERTOOLS14 [52] to confirm simulation stability.

For uniformity, residues comprising solvent, ions, DNA, zinc, and bound effector compounds were stripped from all trajectory and topology files. Additionally, due to modifications at the C182 site in the PK11000-bound mutant and the Y220C mutation itself, residues C182 and C220 (or Y220 in the wild type) were stripped from all trajectories. These steps were performed using the strip and parmstrip commands in AMBERTOOLS [52].

### 4.2. Locally Thresholded Interaction Network Generation

Residue-wise electrostatic and van der Waals (VDW) interaction matrices were computed from atom-wise interaction matrices across regularly sampled frames of each trajectory. For the seven p53-DBD constructs (193 residues), the initial atom-wise interaction matrix (2999 × 2999) was reduced to a residue-level interaction matrix (193 × 193). Edge weights in these matrices represented the total interaction energy between pairs of residues.

To reduce computational cost, the original 1 microsecond trajectory (10,000 frames) was sampled at 66 frames/nanosecond, yielding approximately 150 frames per trajectory. This sampling rate was chosen based on prior studies demonstrating convergence of energetic dynamics using ≥50 sampled frames for smaller systems [30]. Energy computations were performed using the cpptraj utility, and the outputs were parsed into tensors for further analysis. Edge weights were subsequently normalized and thresholded to sparsify the networks, highlighting regions with significant energetic contributions while minimizing less impactful interactions [30,36]. Histogram data of the normalized edge weight values for each construct’s locally-thresholded electrostatic interaction networks and locally-thresholded van der Waals interaction networks can be found in Appendix A respectively.

### 4.3. Heat Kernel Generation from Locally Thresholded Networks

For each p53-DBD construct, the heat kernel was computed from the locally thresholded [36] and normalized electrostatic interaction networks. The heat kernel, a mathematical function modeling the diffusion of information across a network over time, emphasizes node connectivity and topological features at both local and global scales. The heat kernel matrix ht is defined as follows:(1)$$h_{t}=\Phi \left(e^{-t\Lambda }\right)\Phi^{T}$$

Here, Φ is the matrix of spectral eigenvectors of the normalized Laplacian of a graph G, and $\Phi \left(e^{-t\Lambda }\right)\Phi^{T}$ is the diagonal matrix of eigenvalues weighted by the diffusion parameter t.

### 4.4. Determination of the Diffusion Parameter for Heat Kernel Computation

To optimize the diffusion parameter t, we employed a “knee point” detection algorithm [66] based on curvature analysis of eigenvalue distributions [67]. The knee point represents the eigenvalue beyond which additional dimensions add minimal variance, ensuring efficient dimensionality reduction [66]. Using the wild-type p53-DBD construct as a baseline, we identified *t* = 6 as the optimal parameter for all constructs, capturing the majority of variance in the heat kernel’s principal components. Eigenvalue Scree plot data for the diffusion parameter optimization protocol can be found in Appendix A.

### 4.5. Principal Component Analysis (PCA) of Heat Kernels

PCA was performed on the heat kernels to reduce dimensionality and identify significant variances in node connectivity across simulation frames. The mean-centered heat kernel, ${\bar{h}}_{t}$, of each construct was calculated as the average of all 152 frame-specific heat kernels:(2)$${\bar{h}}_{t}=\frac{1}{T}\sum_{f=1}^{T}h_{t_{f}}$$
where $h_{t_{f}}$ is the heat kernel for a specific construct calculated with diffusion parameter value *t* at sampled fame *f*. We hypothesized projecting all sampled frames’ heat kernels across the mean-centered heat kernels three leading eigenvectors would account for the most significant variances in heat kernel matrix values and thus network organization across MD simulation time. Residues’ heat kernel embeddings were projected into a shared latent space using the leading three eigenvectors (PC^1^, PC^2^, and PC^3^) of each construct’s mean centered heat kernel, yielding a total of 29,336 embeddings per construct (193 residues × 152 frames). This embedding provided insights into residue-level network dynamics across simulation time. Data for each construct’s electrostatic heat kernel embedding projections into R^3^ principal component space color-mapped by residue index can be found in Appendix A.

### 4.6. Calculation of Embedding Error Metrics

Embedding error (EE) values were computed using the Wasserstein distance metric, which quantifies the “cost” of transforming one residue’s embedding distribution into another. To calculate the embedding error (EE) for each residue between two p53-DBD constructs, we employed a calculation of the 1st Wasserstein distance metric implemented by the SciPy package [68]. For each residue, EE values were calculated between wild-type, mutant, and drug-bound constructs.

For all residues in the p53-DBD, *i* = 96 to *i* = 290 (excluding *i* = 182 and *i* = 220), let $EE\left(Y220C,Y220C_{d_{j}}\right)$ denote the Wasserstein embedding error value of p53-DBD residue i in the calculated embedding error distribution between construct x and Y220C mutant-bound p53 to drug *dj*. For each DBD residue *i*, we can compute its corresponding value vi in the embedding error difference (EED) distribution *V_j_* of each drug compound *j*, where $v_{i}\in V_{j}$, simply as(3)$$v_{i}=EE{\left(Y220C,Y220C_{d_{j}}\right)}_{i}-EE{\left(wt,Y220C_{d_{j}}\right)}_{i}$$

Thus, when the EE value for residue *i* between the constructs of unbound Y220C and Y220C bound to drug *d_j_* is high and its EE value between WT and Y220C bound to drug *dj* is low, the drug compound redistributes the electrostatic network interactions of residue *i* “toward” its electrostatic network dynamics in nominal wild-type p53 and “away” from its dynamics in the aberrant Y220C mutant construct, resulting in its corresponding EED value being higher. Appendix A visualizes investigated EED cutoff values onto the EED distributions generated for this study.

### 4.7. Code Development

Algorithms implemented in the methodologies were generated in Bash (v5.0, Chet Ramey) and Python (v3.7, Python Software Foundation, Wilmington, DE, USA) [69] through Jupyter Notebooks (v7.0.0, Project Jupyter, Avignon, France) [70]. See supplementary documentation for more information on code implementation. Flow chart diagram of electrostatic network, embedding error, and embedding error difference protocols can be found in Appendix A. 

### 4.8. Molecular and Chemical Structure Visualization

All 3D molecular structures were generated using PyMOL [56], and visualization of chemical structures of small effector compounds and **PK11000** was generated using Ketcher (v3.4.0, EPAM Life Sciences, Newtown, PA, USA) [71].


Appendix A


Sample code can be found in the following files:


Appendix A

Appendix A

Appendix A

Appendix A

Appendix A

Appendix A

Appendix A

Appendix A

Appendix A

Appendix A



Appendix A



Appendix A
Appendix A.

## 5. Conclusions

This study demonstrates the utility of embedding the long-range electrostatic dynamics of the p53 DNA-binding domain (DBD) within a network-theoretic framework to analyze the differential effects of allosteric drug interactions. By integrating heat kernel embedding and Wasserstein distance-based metrics, we identified specific energetic network reorganizations associated with functional, dysfunctional, and drug-rescued conditions of the Y220C mutant p53-DBD. These methodologies revealed the capacity to not only evaluate the efficacy of allosteric drugs but also to pinpoint functionally significant regions across the DBD that are most responsive to drug binding.

Our findings showed that compound **22** from the small-molecule panel was particularly effective in restoring long-range electrostatic interactions in the Y220C mutant, mirroring the restorative effects of the established allosteric modulator PK11000. Importantly, compound **22** achieved this without covalent binding, suggesting its potential as a safer alternative for therapeutic development. Regions such as L1, S6-S7, and L6 emerged as critical sites for electrostatic rescue, highlighting their importance in maintaining p53-DBD stability and function. These findings underscore the potential of targeting distal allosteric sites in the design of drugs capable of inducing global electrostatic network reorganizations toward wild-type-like behavior.

Moreover, this study underscores the broader significance of long-range electrostatics in regulating p53-DBD function and allosteric drug rescue. Our analyses revealed that the dynamic redistribution of electrostatic interactions is a critical determinant of functional state transitions in p53 and likely other proteins with complex allosteric mechanisms. By coupling heat kernel embeddings with residue-specific embedding error analysis, we provide a robust computational framework for dissecting the allosteric effects of small-molecule modulators.

However, our results also emphasize the specificity of allosteric drug interactions. While compound **22** and **PK11000** demonstrated efficacy in rescuing the electrostatic dynamics of the Y220C mutant, the diversity of p53 mutations suggested that no single allosteric drug could universally restore functionality across all variants. Effective therapeutic strategies may require mutation-specific drug designs targeting distinct electrostatic and energetic pathways. As such, the development of tailored allosteric modulators will necessitate detailed analyses of the unique energetic disruptions caused by individual mutations.

Finally, the methodologies developed in this study, including heat kernel embeddings and Wasserstein-based embedding error analyses, offer a powerful toolset for investigating allosteric dynamics in p53 and other proteins. These approaches can aid in both drug discovery and the mechanistic elucidation of allosteric regulation in diverse biological systems. By leveraging these methods, future studies can further explore the therapeutic potential of allosteric modulation in addressing the functional disruptions caused by mutations in p53 and beyond.

## Figures and Tables

**Figure 1 ijms-26-06884-f001:**
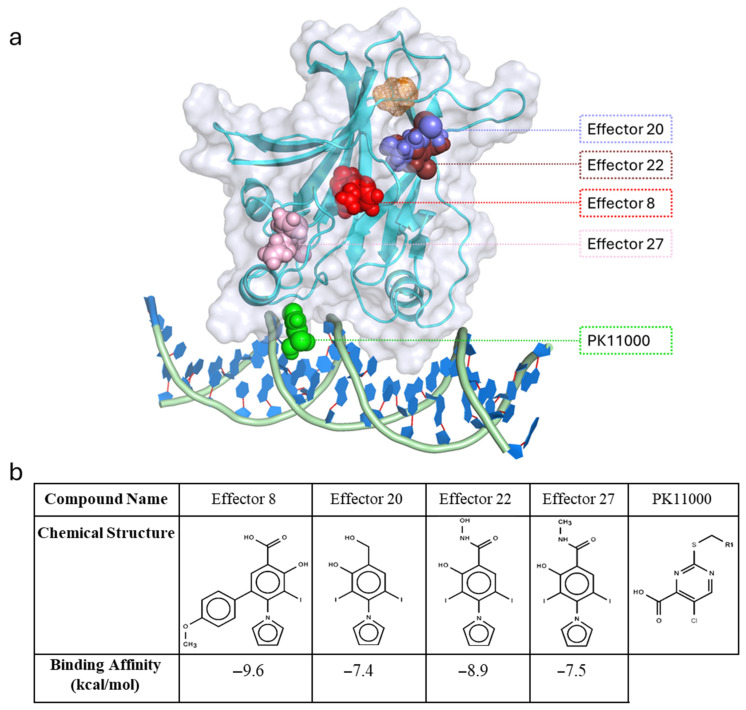
(**a**) Cartoon representation of Y220C mutant p53-DBD bound to DNA and docked effectors. Docked effector compounds are represented in spheres: PK11000 in green, Effector 8 in red, 20 in violet, 22 in brown, 27 in pink, and the Y220C mutant site represented by an orange mesh. (**b**) Table displaying structural data of PK11000 and the four effector compounds, along with their respective binding affinity data [29], that were identified and allosterically docked to the average global structures of 1 us WT p53 and Y220C p53 simulations sourced from Han et al. [30].

**Figure 2 ijms-26-06884-f002:**
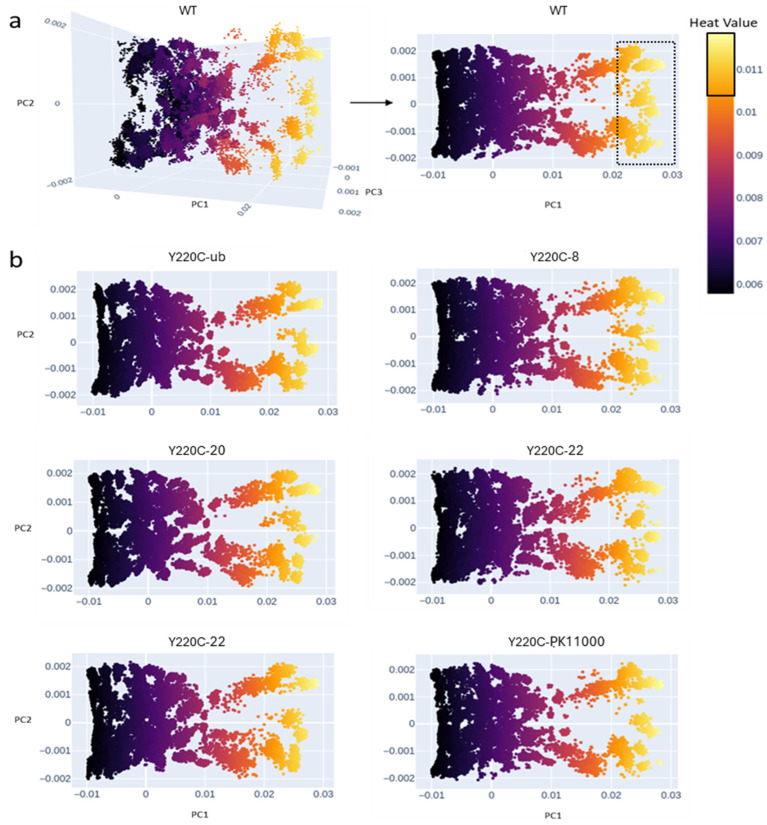
Electrostatic heat kernel principal component (PC) projections of all 193 residues for each of the seven investigated p53 DBD constructs’ 152 sampled frames of simulation. In total, there are 152 × 193 = 29,336 embedded points in R3 principal component space for each construct. (**a**) The projection of the wild-type construct’s 152 heat kernels in R3 across PC^1^, PC^2^, and PC^3^ (left) and R2 across PC1 and PC2 (right) are displayed. (**b**) While calculated in R3, the projections of the remaining six constructs: Y220C, Y220C-8, Y220C-20, Y220C-22, Y220C-27, and Y220C-PK11000 are only shown across R2 for clarity. A Y220C construct bound to the effector drug compound **j** is denoted as **Y220C-j**. Color mapping indicated by the heat value legend represents the degree of node connectivity in the protein system. The more yellow the node embedding, the higher that node embedding’s degree of connectivity. The solid box on the legend indicates ranges of node embedding connectivity values at or above 0.0105. The dotted box surrounding the WT heat kernel demonstrates how the projection of these corresponding high connectivity values distributes across the latent PC space.

**Figure 3 ijms-26-06884-f003:**
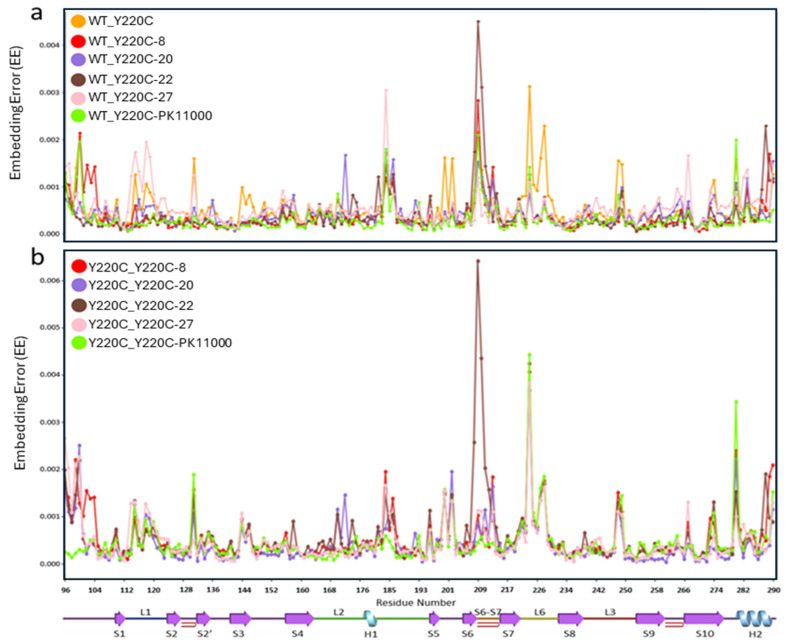
Embedding error analysis. A single 193-residue-long EE distribution consisting of each residue’s Wasserstein EE value calculated between the pair of heat kernel R^3^ projections corresponding to p53-DBD constructs U and V is referred to as U_V. (**a**) The six Wasserstein EE value distributions calculated between the wild type (WT) and each of the six Y220C-mutant constructs: (Y220C-ub, Y220C-27, Y220C-20, Y220C-22, and Y220C-PK11000) across all 193 p53-DBD residues. (**b**) The five Wasserstein EE value distributions calculated between the unbound mutant (Y220C) and each of the five Y220C-mutant drug-bound constructs (Y220C-ub, Y220C-27, Y220C-20, Y220C-22, and Y220C-PK11000) across all 193 p53-DBD residues. The *X*-axis represents the residue number, and the *y*-axis represents the Wasserstein EE value. For both (**a**) and (**b**), Wasserstein EE distributions of either WT_r or Y220C-ub_r where r is Y220C-ub, Y220C-8, Y220C-20, Y220C-22, Y220C-27, or Y220C-PK11000 are colored orange, red, purple, brown, pink, or green, respectively. p53-DBD secondary structural elements corresponding to residue positions are visualized with a line representation under the *x*-axis of (**b**). Loop and helical regions L1 (F113-S121), L2 (K164-P177), H1 (H178-E180), S6-S7 (D207-R213), L6 (E221-T230), L3 (N239-L252), and H2 (D281-R290) illustrate particular areas of note.

**Figure 4 ijms-26-06884-f004:**
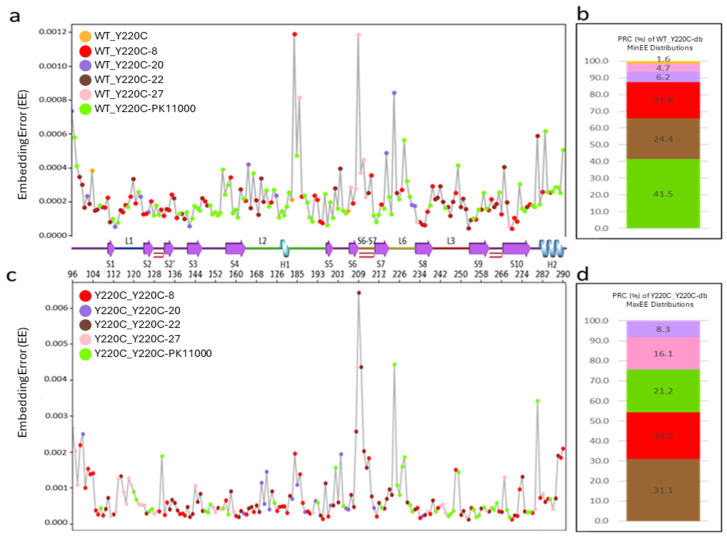
(**a**) Minimal embedding error analysis (MinEEa): a single residue distribution displaying each residue’s lowest EE value from one of the six possible embedding error distributions between the wild type (WT) and a Y220C-construct as generated in Figure 2a. (**b**) The percent residue composition (PRC) of each WT_r EE distribution: The stacked bar graph describes the percentage of all 193 p53-DBD residues whose lowest EE value (as displayed in panel **a**) is consequent of its calculation for each specific WT_r EE distribution. (**c**) Maximal embedding error analysis (MaxEEa): a single residue distribution displaying each residue’s highest EE value from one of the five possible embedding error distributions between Y220C and a Y220C-db construct. The *X*-axis represents the residue number, and the *y*-axis represents the EE value. (**d**) The PRC for each Y220C_Y220C-db EE distribution: the stacked bar graph describes the percentage of all 193 p53-DBD residues whose highest embedding error value (as displayed in **c**) is consequent of its calculation from each specific Y220C_Y220C-db EE distribution. Coloration follows from Figure 3. P53-DBD secondary structural elements corresponding to residue number are visualized with a line representation under the *x*-axis.

**Figure 5 ijms-26-06884-f005:**
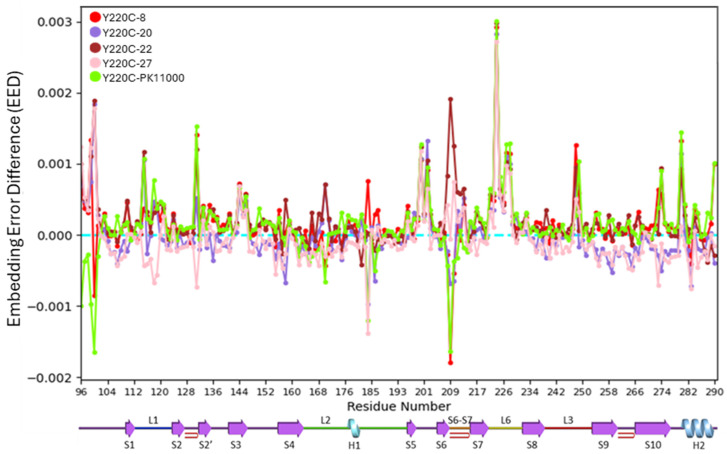
Embedding error difference (EED) distributions for each of the five Y220-db constructs relative to the WT and Y220C constructs across all 193 DBD residues. The *X*-axis corresponds to the residue index, and the *Y*-axis corresponds to the EEE value calculated as the difference (error) between each residue’s EE value from the Y220C-ub_Y220C-j and WT_Y220C-j distributions, where j is one of the five examined drug compounds: **8**, **20**, **22**, **27**, and **PK11000**. A blue dotted line is drawn across the *x*-axis to indicate an EEE value of 0. Secondary structural elements of the p53-DBD are visualized under the *x*-axis. Coloration as dependent on construct Y220C-j follows from Figure 3.

**Figure 6 ijms-26-06884-f006:**
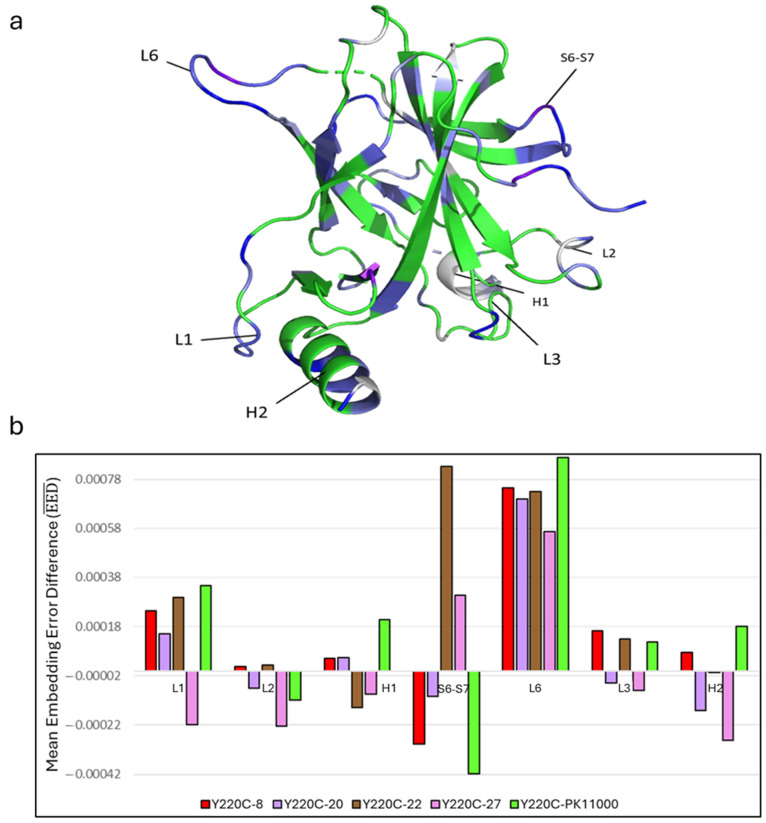
(**a**) Residues with EED values greater than or equal to 0.0002 mapped onto the structure of p53’s DBD from any of the five EED distributions. Color mapping of white-to-blue-to-purple corresponds to residues with increasing EED values greater than or equal to 0.0002, 0.00025, 0.0003, 0.0004, 0.0005, 0.001, and 0.002. (**b**) Grouped bar chart of mean EED value ($\overset{―}{\mathrm{EED}}$) from each of the five EED distributions averaged across residues in functionally implicated loop and helical regions L1, L2, H1, S6-S7, L6, L3, and H2. Coloration follows from those assigned to each Y220C-db construct in Figure 3.

**Table 1 ijms-26-06884-t001:** Tabulation of percentages of residues in functionally relevant loop and helical motif regions with EED values in increasingly high threshold ranges. The investigated loop and helical motif regions include L1 (F113-S121), L2 (K164-P177), H1 (H178-E180), S6-S7 (D207-R213), L6 (E221-T230), L3 (N239-L252), and H2 (D281-R290). The more green-shifted the color mapping, the greater the residue percentage. The preceding column indicates the average alpha-carbon distance of the region from Y220C. The more red-shifted the color mapping, the more proximal the motif is to the Y220C mutation site.

		% of Residues with EED Value ≥ Cuttoff					
		Residue EED Value Cutoff					
p53-DBD Motif	Average α-carbon distance (Å) between Motif and Y220C	0.0002	0.0003	0.0004	0.0005	0.001	0.002
L1	29.1	54.5	45.5	36.4	18.2	9.1	0.0
L2	25.5	38.5	7.7	3.8	0.0	0.0	0.0
H1	29.1	66.7	0.0	0.0	0.0	0.0	0.0
S6-S7	19.4	85.7	71.4	71.4	71.4	28.6	14.3
L6	15.2	90.0	70.0	70.0	70.0	30.0	10.0
L3	30.5	28.6	21.4	14.3	14.3	14.3	0.0
H2	36.7	70.0	46.2	30.8	15.4	15.4	0.0

## Data Availability

The data presented in this study are available on request from the corresponding author. The data are not publicly available due to insufficient storage size for distributing the raw trajectory file data utilized in the present study.

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
