# Peer review of "Network Theory Analysis of Allosteric Drug-Rescue Mechanisms in the Tumor Suppressor Protein p53 Y220C Mutant"

_ijms, 2025, doi:10.3390/ijms26146884_

Round 1
Reviewer 1 Report
Comments and Suggestions for Authors
Review of the manuscript “Network Theory Analysis of Allosteric Drug-Rescue Mechanisms in the Tumor Suppressor Protein p53 Y220C Mutants”
In this article, the authors use heat kernel and Wasserstein distance-based analyses to study the allosteric effect that several drugs have on the p53 protein in the wild type form and Y220C mutant. Using this network theory-based protocol, the authors demonstrate that changes in electrostatic interactions can be used to measure the allosteric effect of small drugs. Overall, the article presents a novel approach to study the allosteric effect of drugs. The following comments should be addressed before publication.
Comments:
- A figure is needed to show the chemical structure of the allosteric drugs, the allosteric binding site, and the location of the Y220C mutation.
- In Figure 1, it is difficult to see the differences between the electrostatic heat kernel PC projections. Is there a way to quantify how different these are upon binding of the different molecules?
- Is the DNA included in the simulations? The authors do not describe the force field for the DNA but then they mention that residues comprising DNA were stripped. If DNA was not included in the simulations, is the protein stable without it?
- How is the C183-pk1100 residue parametrized?
- The authors mention: “The sampling rate was chosen based on prior studies demonstrating convergence of energetic dynamics…” However, they do not cite the previous study.
- Is there a way to validate their predictions regarding the efficacy of compound 22 in rescuing the electrostatic dynamics?
Minor comments
- In the title, “mutants” should be mutant
- Some references are not in superscript
- WT and Y220C appear not capitalized on many occasions
Author Response
Reviewer 1:
- A figure is needed to show the chemical structure of the allosteric drugs, the allosteric binding site, and the location of the Y220C mutation.
We thank the reviewer for this helpful suggestion to include an orienting figure. We note that Reviewer 2 also requested that Supplementary Figure 1 be moved into the main manuscript, and we believe that revision also addresses this concern. We have reworked the figure to more clearly highlight the chemical structures of the allosteric compounds, the location of the Y220C mutation, and the allosteric binding site. The revised figure now appears as Figure 1 in the main text and provides a structural overview to help readers contextualize the drug-rescue mechanism.
- In Figure 1, it is difficult to see the differences between the electrostatic heat kernel PC projections. Is there a way to quantify how different these are upon binding of the different molecules?
We thank the reviewer for their observation. While one can not quantitatively compare the figures by eye, the procedure outlined in our Supplementary Figure S6, particularly panel e, explains how we carried out such analysis. The quantitative differences are presented in Figure 3.
- Is the DNA included in the simulations? The authors do not describe the force field for the DNA but then they mention that residues comprising DNA were stripped. If DNA was not included in the simulations, is the protein stable without it?
Yes, we have reported on simulations including the DNA. To make this point more clear, we have noted it in the Methods section, including the force field information, and provided a clear citation to the detailed original report on these simulations.
“The MD simulation trajectory data for the unbound wildtype p53-DBD construct, the unbound Y220C p53-DBD mutant construct, and the five mutant Y220C p53-DBD constructs bound to effector compounds (pk11000, 8, 20, 22, and 27) were sourced from Han et al.’s recently published MD docking and MD-MSM study on the Y220C p53 mutant13.
All-atom MD simulations for each p53-DBD construct (residues 96–290) were conducted for 1 microsecond (10,000 simulation frames) using explicit solvent models, water, DNA, and counterions. Simulations followed a standard protocol developed by the Thayer lab, employing the AMBER14 and AMBER16 packages with the AMBERTOOLS14 suite 51-53. The FF19SB force field 53, 54 was used for proteins, and the TIP3P potential was applied for solvent modeling.30”
- Han, I. S. M.; Abramson, D.; Thayer, K. M. Insights into Rational Design of a New Class of Allosteric Effectors with Molecular Dynamics Markov State Models and Network Theory. ACS Omega 2022, 7(24), 2831–2841. doi:10.1021/acsomega.1c05624 pmc.ncbi.nlm.nih.gov+8
- How is the C183-pk1100 residue parametrized?
We appreciate the reviewer’s attention to the parameterization of this covalently bound system. The C183 residue, which forms a covalent bond with PK11000, was modeled using the standard CYX residue type within the AMBER force field. This residue designation accounts for the sulfhydryl linkage to form covalent bonds, and is accessed during system preparation by modifying the residue name from CYS to CYX in the tleap input, enabling appropriate force field parameters for bonding.
For PK11000, a non-standard small molecule, we used the Generalized Amber Force Field (GAFF), implemented via Antechamber in the AMBER suite. This process includes a quantum mechanical geometry optimization and calculation of partial atomic charges, alongside a set of parameters suitable for use with small molecules not included in the default force field libraries. It is possible to treat the linkage of small molecules to a residue through setting up bondable atom types in the residue. The cross linking occurs during setup with tleap. This is a widely accepted method in the molecular dynamics community for modeling small molecules.
To improve clarity, we have added a citation to Han et al., where our initial simulations of PK11000 were presented, and we now explicitly cite the use of GAFF in the revised manuscript:
“PK11000 was parameterized using the Generalized Amber Force Field (GAFF)30, implemented via Antechamber in the AMBER suite56 and covalently bonded to C182 using the t-Leap program...”
Citations:
# original simulation paper
- Han, I. S. M.; Abramson, D.; Thayer, K. M. Insights into Rational Design of a New Class of Allosteric Effectors with Molecular Dynamics Markov State Models and Network Theory. ACS Omega 2022, 7(24), 2831–2841. doi:10.1021/acsomega.1c05624 pmc.ncbi.nlm.nih.gov+8
# docking study which also included PK11000
- Han, S. M.; Thayer, K. M. Reconnaissance of Allostery via the Restoration of Native p53 DNA‑Binding Domain Dynamics in Y220C Mutant p53 Tumor Suppressor Protein. ACS Omega 2024, 9 (18), 19837–19847. wuxibiology.com+9pubmed.ncbi.nlm.nih.gov+9researchgate.net+9
# paper explaining GAFF
- General AMBER Force Field (GAFF) citation:
- Wang, J.; Wolf, R. M.; Caldwell, J. W.; Kollman, P. A.; Case, D. A. Development and testing of a general AMBER force field. J. Comput. Chem. 2004, 25(9), 1157–1174.
- The authors mention: “The sampling rate was chosen based on prior studies demonstrating convergence of energetic dynamics…” However, they do not cite the previous study.
Thank you for mentioning that we did not explicitly cite this. We have now added the citation, which is our lab’s same paper as noted in the item above.
“The sampling rate was chosen based on prior studies30 demonstrating…”
- Han, I. S. M.; Abramson, D.; Thayer, K. M. Insights into Rational Design of a New Class of Allosteric Effectors with Molecular Dynamics Markov State Models and Network Theory. ACS Omega 2022, 7(24), 2831–2841. doi:10.1021/acsomega.1c05624 pmc.ncbi.nlm.nih.gov+8
- Is there a way to validate their predictions regarding the efficacy of compound 22 in rescuing the electrostatic dynamics?
We thank the reviewer for this question. The compounds analyzed in this study, including compound 22, were selected specifically because their efficacy in rescuing the dynamics and function of mutant p53 has been previously demonstrated in reference #13. Our goal in the present work is to provide mechanistic insight into how these compounds achieve that effect, particularly through changes in electrostatic signaling networks. To clarify this, we have added discussion and citation in the revised manuscript to explicitly state the basis for compound selection and the experimental validation that motivated this analysis.
- Han, S. M.; Thayer, K. M. Reconnaissance of Allostery via the Restoration of Native p53 DNA‑Binding Domain Dynamics in Y220C Mutant p53 Tumor Suppressor Protein. ACS Omega 2024, 9 (18), 19837–19847. wuxibiology.com+9pubmed.ncbi.nlm.nih.gov+9researchgate.net+9
Minor comments
- In the title, “mutants” should be mutant
Thank you for pointing this out. We have fixed this typo.
- Some references are not in superscript
Thanks for your careful eye. We have rerun our citation manager which has now consistently represented all the citation links.
- WT and Y220C appear not capitalized on many occasions
Thanks for noting our capitalization was not carried out in a consistent manner. We have used search and replace to standardize these throughout.

Reviewer 2 Report
Comments and Suggestions for Authors
Title: Network Theory Analysis of Allosteric Drug-Rescue Mechanisms in the Tumor Suppressor Protein p53 Y220C Mutants
The work provides insights into the allosteric modulation of various allosteric modulators of the p53-DBD protein, utilizing heat kernel and Wasserstein distance-based analyses to study the long-range electrostatic dynamics.
Comments:
- Figure 1: It would be beneficial for the authors to highlight the regions of discussion, specifically “heat kernel values ≥ 0.015” and “heat kernel values ≤ 0.0095”, with dashed boxes. This will help readers focus and compare these regions across samples.
- Results - Energetic Network Generation and Electrostatic Heat Kernel PCA Projection:
- In line 153, the use of the word “subtle” raises the question of what defines “subtle”. What is the error associated with the analysis? How do we determine which changes in the heat map are significant or not?
- Figure 2:
- Remove the “wt” from the legends in Figure 2a. It is not accurate to include wild-type plus mutation in the same name.
- Additionally, it would be helpful to highlight the region of interest in the plots to guide the reader.
- What defines the significant change in EE values? What is the error associated with the experiment?
- Figure 3:
- Ensure the naming of samples is consistent throughout the figures.
- Figure 4:
- There are no legends on the plot. Please add legends.
- Adding a schematic to show the summary of positive EED values promoting the restorative impact towards wild-type and negative EED values suggesting destabilization of the electrostatic network towards unbound Y220C would be impactful.
- Figure 5:
- I suggest the reviewer add a structural figure showing the different important regions and highlighting the binding pocket for drugs and the site of mutation. This would help the reader relate the findings to the structure.
- What is the PDB ID for the structure? Please add this information.
- It would also be better to combine Figure 5 and Figure 6 in a complementary manner to highlight key findings and summarize the results.
- Figure 6:
- Add the y-axis label.
General Comments:
- How do these findings contribute to exploring the mechanistic understanding of the allosteric drug rescue mechanism?
- What are the other relevant mutations that can be explored with a similar approach? It would be beneficial to expand the exploration to other critical mutations to understand the mechanism.
- It would be better to add Supp. Fig.1 as the main Figure 1.
Author Response
Reviewer 2:
Comments:
- Figure 1: It would be beneficial for the authors to highlight the regions of discussion, specifically “heat kernel values ≥ 0.015” and “heat kernel values ≤ 0.0095”, with dashed boxes. This will help readers focus and compare these regions across samples.
Thank you for this note. We have indicated the shade cutoff on the colorbar, as well as on the figure.
- Results - Energetic Network Generation and Electrostatic Heat Kernel PCA Projection:
- In line 153, the use of the word “subtle” raises the question of what defines “subtle”. What is the error associated with the analysis? How do we determine which changes in the heat map are significant or not?
We agree that the use of the word subtle does raise questions of definition. To avoid this, we have removed the word, and allowed the data to speak for itself in terms of interpretations.
- Figure 2:
- Remove the “wt” from the legends in Figure 2a. It is not accurate to include wild-type plus mutation in the same name.
Please allow us to explain why we have used the seemingly contradictory notation. The reason is that the data must be computed with respect to a reference. To emphasize this, our naming convention utilizes two fields, the first of which explicilty calls out the reference state, which is the wt state. This is consistent with the nomenclature used throughout the manuscript to discuss any aspects where 0th order referencing is involved. For these reasons, we prefer to leave the legends in Figure 2a as we have them.
- Additionally, it would be helpful to highlight the region of interest in the plots to guide the reader.
This is a great suggestion. However, we felt that making further annotations with boxes on an already very busy figure detracted from the readability of the data. What we have done instead is noted the areas of interest, which correspond to the structural annotations along the X-axis in the figure, in the figure caption.
lines 209 -211:
“Figure 3… p53-DBD secondary structural elements corresponding to residue positions are visualized with a line representation under the x-axis of (b). Regions L1, H1, S6-S7, L6, L3, and H2, illustrate particular areas of note.”
- What defines the significant change in EE values? What is the error associated with the experiment?
We apologize for the confusion about this naming convention and thank the reviewer for pointing this out. The name of the method is “embedding error analysis” but it may be more aptly thought of as an “embedding difference analysis.” Our decision on the naming scheme follows naming conventions of prior literature on the topic of distance metrics, reference number 27.
- Cowan, B. S.; Beveridge, D. L.; Thayer, K. M., Allosteric Signaling in PDZ Energetic Networks: Embedding Error Analysis. The Journal of Physical Chemistry B 2023.
- Figure 3:
- Ensure the naming of samples is consistent throughout the figures.
Thank you, we have compared across figures to ensure consistent labeling.
- Figure 4:
- There are no legends on the plot. Please add legends.
We have added legends; thank you for pointing it out.
- Adding a schematic to show the summary of positive EED values promoting the restorative impact towards wild-type and negative EED values suggesting destabilization of the electrostatic network towards unbound Y220C would be impactful.
We have added Supplementary Figure 6, a schematic flow chart explaining the procedure.
- Figure 5:
- I suggest the reviewer add a structural figure showing the different important regions and highlighting the binding pocket for drugs and the site of mutation. This would help the reader relate the findings to the structure.
Thank you, there was a similar suggestion by reviewer 1, and this is similar to your suggestion of moving the supplementary figure to the main paper; we have done this and labeled the pertinent structural features disussed in this paper.
- What is the PDB ID for the structure? Please add this information.
No PDB ID is reported because the structure was rendered from a snapshot taken from our MD simlations.
- It would also be better to combine Figure 5 and Figure 6 in a complementary manner to highlight key findings and summarize the results.
Thank you, we have merged the figures.
- Figure 6:
- Add the y-axis label.
Thank you, we added the y-axis label on what is now Figure 5B.
General Comments:
- How do these findings contribute to exploring the mechanistic understanding of the allosteric drug rescue mechanism?
We appreciate the reviewer’s interest in the mechanistic implications of our findings. The goal of this study is precisely to uncover the underlying signaling pathways through which rescue compounds restore function to the Y220C mutant of p53. By using heat kernel-based network analysis and electrostatic flow mapping, we identify shifts in long-range communication patterns that propagate from the ligand-binding site toward functional regions of the protein. These shifts are not evident from structural data alone but emerge clearly in the latent space of electrostatic dynamics. Our results suggest that the rescue mechanism operates by re-establishing disrupted electrostatic pathways, enabling more native-like residue-residue interactions that are critical for restoring wild-type behavior. We have additionally revised our manuscript to address this throughout the Discussion section.
- What are the other relevant mutations that can be explored with a similar approach? It would be beneficial to expand the exploration to other critical mutations to understand the mechanism.
We thank the reviewer for this insightful and forward-looking question. We agree that this approach can be extended to study additional hotspot mutations in p53, many of which are of high biological and clinical interest. More broadly, the network-based framework we present is readily generalizable to any residue of interest in systems for which molecular dynamics simulations are available. We view this method as a flexible tool with broad applicability for uncovering dynamic and allosteric consequences of mutations across a range of protein systems. We have added a brief statement to the Discussion to underscore the potential for future applications:
lines 509-517:
Our approach can be extended to study additional hotspot mutations in p53, many of which are of high biological and clinical interest due to their prevelence in. In order of respective prevelance in p53-DBD, R175H, R248Q, R273H, R248W, R175L, R273C, R282W, R248L, and R175P are of particular interest with the first six of these mutations being accountable for 30% of all p53-DBD oncogenic mutations51. More broadly, the network-based framework we present is readily generalizable to any residue of interest in systems for which MD-simulations are available. We view this method as a flexible tool with broad applicability for uncovering dynamic and allosteric consequences of mutations across a range of protein systems.
- Thayer, K.M., Stetson, S., Caballero, F. et al.Navigating the complexity of p53-DNA binding: implications for cancer therapy. Biophys Rev16, 479–496 (2024). https://doi.org/10.1007/s12551-024-01207-4
- It would be better to add Supp. Fig.1 as the main Figure 1.
Thank you for suggesting this. We found this useful to address your concern about adding a figure with structural infomration, which was similar to a request by reviewer 1 as well. Please also see above.
We thank the reviewers and editors once again for their thoughtful comments and suggestions, which have helped us to improve the clarity and presentation of our work. We are enthusiastic about the opportunity to share these findings with the broader scientific community, and we hope that the revised manuscript will now be found suitable for publication in International Journal of Molecular Sciences. We look forward to the possibility of bringing this article to publication and contributing to the important ongoing conversations in the field.
